Study on pyroptosis-related genes Casp8, Gsdmd and Trem2 in mice with cerebral infarction

Liang Shunli 1 2
Xu Linsheng 1
Xin Xilin 1
Zhang Rongbo 1
Wu You 20071023@zcmu.edu.cn 1
1 Department of Neurology, The Second Affiliated Hospital of Zhejiang Chinese Medical University , Hangzhou , Zhejiang , China
2 The Second Clinical Medical College of Zhejiang Chinese Medical University , Hangzhou , Zhejiang , China
Uversky Vladimir
Electronic publication date: 2024 Feb 9
Publication date: 2024
Volume: 12
Electronic Location ID: e16818
Received 2023 Sep 8; Accepted 2023 Dec 30
Copyright: ©2024 Liang et al.
Copyright year: 2024
Copyright holder: Liang et al.
License: This is an open access article distributed under the terms of the Creative Commons Attribution License, which permits unrestricted use, distribution, reproduction and adaptation in any medium and for any purpose provided that it is properly attributed. For attribution, the original author(s), title, publication source (PeerJ) and either DOI or URL of the article must be cited.
License URL: https://creativecommons.org/licenses/by/4.0/

Keywords: Cerebral infarction, Bioinformatics, Cell pyroptosis gene, Middle cerebral artery occlusion model, Pyroptosis genetic verification

Funding: Zhejiang Outstanding Young Talent Fund of Traditional Chinese Medicine 2021ZQ043 This work was supported by the Zhejiang Outstanding Young Talent Fund of Traditional Chinese Medicine (No. 2021ZQ043). The funders had no role in study design, data collection and analysis, decision to publish, or preparation of the manuscript.

==============================
Objective

Cerebral infarction is the main cause of death in patients with cerebrovascular diseases. Our research aimed to screen and validate pyroptosis-related genes in cerebral infarction for the targeted therapy of cerebral infarction.

Methods and results

A total of 1,517 differentially expressed genes (DEGs) were obtained by DESeq2 software analysis. Gene set enrichment analysis results indicated that genes of middle cerebral artery occlusion (MCAO) mice aged 3 months and 18 months were enriched in pyroptosis, respectively. Differentially expressed pyroptosis-related genes (including Aim2, Casp8, Gsdmd, Naip2, Naip5, Naip6 and Trem2) were obtained through intersection of DEGs and genes from pyroptosis Gene Ontology Term (GO:0070269), and they were up-regulated in the brain tissues of MCAO mice in GSE137482. In addition, Casp8, Gsdmd, and Trem2 were verified to be significantly up-regulated in MCAO mice in GSE93376. The evaluation of neurologic function and triphenyltetrazolium chloride staining showed that the MCAO mouse models were successfully constructed. Meanwhile, the expressions of TNF-α, pyroptosis-related proteins, Casp8, Gsdmd and Trem2 in MCAO mice were significantly up-regulated. We selected Trem2 for subsequent functional analysis. OGD treatment of BV2 cell in vitro significantly upregulated the expressions of Trem2. Subsequent downregulation of Trem2 expression in OGD-BV2 cells further increased the level of pyroptosis. Therefore, Trem2 is a protective factor regulating pyroptosis, thus influencing the progression of cerebral infarction.

Conclusions

Casp8, Gsdmd and Trem2 can regulate pyroptosis, thus affecting cerebral infarction.

Introduction

Cerebral infarction is a serious cerebrovascular disease. It is a neurological disease with recurrence (Memon et al., 2021). Cerebral infarction is associated with risk factors such as unhealthy lifestyles and diabetes (Goldstein et al., 2011). It has been universally known that the rapid metabolism of the brain requires a large amount of oxygen, and brain cells die due to hypoxia leading to cerebral infarction (Wakhloo et al., 2020). At the same time, studies have shown that cerebral infarction often occurs in the neck or after cerebral blood flow occlusion (Wang et al., 2019). Nowadays, thrombolytic therapy is regarded as an effective way to treat mild cerebral infarction, but most of the patients diagnosed with cerebral infarction are already in critical condition, and the effect of thrombolytic therapy is not satisfactory (Powers et al., 2018; Smith et al., 2019). Therefore, new therapeutic targets for cerebral infarction are urgently needed in clinical practice.

Pyroptosis is a mode of programmed cell death, which contributes to chromatin fragmentation, and finally leads to cell death (Fang et al., 2020). Pyroptosis is an immune defense response of the body, and has been revealed by studies to play a significant role in diseases such as diabetes, diabetic cardiomyopathy, diabetic nephropathy, and myocardial ischemia (Jorgensen & Miao, 2015; Qiu et al., 2017; Han et al., 2018). Pyroptosis mediates renal ischemia injury (Yang et al., 2014; Wu et al., 2016). Neurons, microglia, astrocytes and other brain tissue cells are all dependent on caspase-1 activation (Adamczak et al., 2014; Tan et al., 2015; Barrington, Lemarchand & Allan, 2017). At the same time, cell pyroptosis in cerebral infarction can be activated through the inflammasome pathway, and cell pyroptosis can be an important potential target of cerebral infarction (Tang & Deng, 2018). In conclusion, it was demonstrated that pyroptosis can be a latent therapeutic target for cerebral infarction.

In this research, bioinformatics methods were utilized to screen pyroptosis-related genes as potential therapeutic targets for cerebral infarction. DESeq2 software was applied to analyze GSE137482 data set to obtain differentially expressed genes (DEGs). Furthermore, pyroptosis-related genes were obtained using Venn plot software. Finally, in order to guarantee the reliability and accuracy of results obtained through bioinformatics methods, the expression of the screened pyroptosis-related genes was verified in the brain tissues of mice with cerebral infarction through animal experiments. In conclusion, we provide new targets for patients with cerebral infarction.

Methods

Data acquisition

GSE137482 and GSE93376 were downloaded from the GEO. RNA-SEQ of the brain tissues from 24 mice were included in GSE137482: including 12 mice in the middle cerebral artery occlusion model group (MCAO) and 12 mice in the sham operation group (Sham). There were the brain tissues of six mice in GSE93376: three MCAO models and three sham operation samples. Finally, a total of 30 samples (15 MCAO models and 15 sham operation samples) were used for bioinformatics analysis.

DEGs analysis

The DEGs between the Sham group and the MACO group (mice aged 3 and 18 months) were comparatively analyzed using the DESeq2 software (—logFC—>1, padj <0.05) in GSE137482. Next, Venn diagram analysis was performed on the down- and up-regulated genes from mice aged 3 and 18 months (Fig. 1C and 1D), respectively, and the common DEGs were obtained.

Figure 1 Screening of DEGs.

(A) Volcano map of DEGs in 3-month-old mice; (B) volcano map of DEGs in 18-month-old mice; (C) Venn diagram of up-regulated DEGs; (D) Venn diagram of down-regulated DEGs.

GO and KEGG analysis

ClusterProfiler software is a general enrichment tool for interpreting omics data (Yu et al., 2012; Wu et al., 2021), and we used it to analyze the GO and KEGG enrichment pathways of the DEGs.

GSEA

GSEA was performed on mice aged 3 and 18 months using the GSEA 4.1.0 software.

Identification of Pyroptosis-related Genes

Thirty-five genes in the pyroptosis Gene Ontology Term (GO:0070269) was downloaded from the GO database (Table S1). The Venn plot software was used for the intersection of 35 genes in GO:0070269 and DEGs in GSE137482 to obtain distinctively expressed pyroptosis-related genes. Simultaneously, the PPI network analysis of the pyroptosis-related genes was performed using STRING. Finally, the distinctively expressed pyroptosis-related genes associated with cerebral infarction were confirmed.

Establishment of Mouse MCAO Model

Twenty C57BL/6J mice (23–25 g), aged 8–10 weeks (Nanjing Institute of Biomedical Sciences, Nanjing University), divided into two groups according to the random number table method: the MCAO model group and the sham operation group (n = 6). Finally, a total of 12 samples (6 MCAO models and 6 sham operation samples) were used for experimental verification. The mice were kept in an SPF grade experimental animal room with a 12-hour cycle light system, a humidity of 50 ± 5%, and a temperature of 24 ± 2 °C. Before the experiment, the mice were fed adaptively for one week, and had free eating and drinking during this period. The mouse MCAO model was constructed according to the Longa method (Longa et al., 1989), and the successful establishment of the MCAO model was judged by evaluating the neurological function of the mice. The sham operation group served as the control group, and mice in this group underwent the same surgical procedures, excluding middle cerebral artery occlusion. Neurological examination is divided into 5 levels: 0 points: normal; 1 point: The left front paw cannot be fully extended; 2 points: When walking, rotate the mouse towards the paralyzed side; 3 points: When walking, the mouse body tilts towards the paralyzed side; 4 points: Unable to walk independently and lose consciousness. The mice with a score of 1–4 were included in the MCAO model group, and mice with a score of 0 were enrolled in the sham operation group. After the experiment, the mice were put into a clean container, where carbon dioxide was slowly introduced, reaching a final concentration of 60%. The incremental rise in carbon dioxide concentration induced painless euthanasia of the mice. The animal experiments in this research were approved by the Animal Care and Use Committee (IACUC) of the Laboratory Animals Committee of Zhejiang Province (ZJCLA) (approval number: ZJCLA-IACUC-20050029).

Cell culture and grouping

Mouse microglia BV-2 (CL-0493B; Pricella) were cultured in high-glucose DMEM medium (12491015; Thermo Fisher Scientific, Waltham, MA, USA) containing 10% FBS in a 5% CO2, 37 °C incubator. The cells were exposed to oxygen-glucose deprivation (OGD) for cell model preparation. Cells in the OGD group were cultured with sugar-free DMEM medium (ThermoFisher, #12491015) in a hypoxic incubator containing a pre-mix of gases (94% N2, 1% O2, 5% CO2) at 37 °C. Control group cells cultured under normal conditions (95% air, 5% CO2).

Interference with Trem2 expression

BV2 cells were seeded into 12-well plates at a density of 1. 4 ×105/well, and then incubated with jetPRIME® transfection reagent (Polyplus, #101000001) and siRNA oligonucleotide of Trem2 (Sangon Biotech, Shanghai, China) for 6 h, followed by a change of DMEM culture medium after 24 h. si-NC was constructed as a negative control.

TTC Staining

Twelve mice were euthanized after 3 days, and their brain tissues were obtained after dissection. Cerebral infarction was identified by TTC staining. The brain tissue sections from the MCAO group were prepared and incubated in 1% TTC solution. The slices were turned over to ensure even staining. Finally, organize fixation with 4% paraformaldehyde and photographed for image collection.

qRT-PCR Assay

Total RNA was obtained from the brain tissues of mice using the Trizol kit (Invitrogen, Ca, USA). cDNA was further synthesized and qRT-PCR was performed. The gene expression levels of Casp8, Gsdmd and Trem2 were calculated by the 2−ΔΔCT method, and the calculation was repeated three times.

Primer sequence: Casp8 Forward primer (F): 5′-TGCCCTCAAGTTCCTGTGCTTGGAC-3′, Reverse primer (R): 5′-GGATGCTAAGAATGTCATCTCC-3′; Gsdmd F: 5′-GTGCCTCCACAACTTCCTGA-3′, R: 5′-GTCTCCACCTCTGCCCGTAG-3′; Trem2 F: 5′-CTACCAGTGTCAGAGTCTCCGA-3′, R: 5′-CCTCGAAACTCGATGACTCCTC-3′; β-actin F: 5′-CTAGGCACCAGGGTGTGAT-3′, R: 5′-TGCCAGATCTTCTCCATGTC-3′.

Western Blot Assay

The brain tissues of mice were collected and lysed with RIPA solution. After electrophoresis, the samples were transferred to PDVF membrane, sealed with 5% skim milk for 1 h, and reacted with primary antibody (NLRP3, 1:1000, Casp8, 1:1000, Gsdmd, 1:1000, Trem2, 1:1000, Cleaved caspase-1, 1:1000, GAPDH, 1:1000, Cell Signaling Technology, USA) at 4 °C overnight. After cleaning, the samples were reacted with goat anti-mice IgG (1:3000; Cell Signaling Technology, Danvers, MA, USA) at room temperature for 1 h.

ELISA

After collecting the cell culture supernatant, the expression levels of IL-6, TNF-α and IL-1β were detected by ELISA kit. For details, see the kit instructions.

Statistical analysis

IBM SPSS Statistics 27 was applied for student’s t-test, and P < 0.05 was considered statistically remarkable.

Results

Screening of DEGs

In this study, the DESeq2 software was used to analyze the DEGs of 3-month-old mice between the MACO group and the Sham group in GSE137482, and 1,844 DEGs were found, including 125 down-regulated and 1,719 up-regulated DEGs (Fig. 1A). We comparatively analyzed the DEGs of 18-month-old mice between the MACO group and the Sham group, and 2,527 DEGs were screened out, including 337 down-regulated and 2,190 up-regulated DEGs (Fig. 1B). Meanwhile, Venn diagram analysis showed that there were total 34 down-regulated and 1,483 up-regulated DEGs by intersection of DEGs from mice aged 3 and 18 months (Fig. 1C and 1D).

GO and KEGG

In order to explore the potential biological functions of DEGs, GO and KEGG analyses were performed on the up- and down-regulated DEGs. The up-regulated DEGs were revealed to be enriched in pathways of Human T-cell Leukemia virus 1 infection, T cell activation, positive regulation of cytokine production, Epstein-Barr virus infection, and ECM-receptor interaction (Fig. 2A and 2B). At the same time, down-regulated DEGs were found to be enriched in positive regulation of camp-mediated signaling, cellular response to interleukin −1, and Apelin signaling (Fig. 2C and 2D).

Figure 2 Functional Analysis of DEGs.

(A) GO analysis of the up-regulated DEGs; (B) KEGG analysis of the up-regulated DEGs; (C) GO analysis of the down-regulated DEGs; D: KEGG analysis of the down-regulated DEGs.

GSEA

In addition, for further investigate the functions and effects of DEGs, the GSEA 4.1.0 software was used to conduct GSEA on genes. GSEA showed that genes from MCAO mice aged 3 and 18 months were significantly enriched in pyroptosis (Fig. 3A and 3B).

Figure 3 GSEA.

(A) GSEA of mice aged 3 months; (B) GSEA of mice aged 18 months.

Identification of differentially expressed pyroptosis-related genes

To identify the distinctively expressed pyroptosis-related genes, Venn plot software was used to analyze up-regulated DEGs in GSE137482 and 35 genes in the pyroptosis Gene Ontology Term (GO:0070269), and seven distinctively expressed pyroptosis-related genes were obtained accordingly (Fig. 4A). Meanwhile, a PPI network with 15 edges and seven nodes was obtained (Fig. 4B). In addition, these seven genes (Aim2, Casp8, Gsdmd, Naip2, Naip5, Naip6 and Trem2) were up-regulated in MCAO mice aged 3 and 18 months (Fig. 4C and 4D).

Figure 4 Identification of Differentially Expressed Pyroptosis-related Genes.

(A) Venn plot for the intersection of pyroptosis-related genes and DEGs; (B) a PPI network of seven differentially expressed pyroptosis-related genes; (C) expressions of the differentially expressed pyroptosis-related genes in the brain tissues of mice aged 3 months; (D) differential expression of the pyroptosis-related genes in the brain tissues of 18-month-old mice.

Verification of differentially expressed pyroptosis-related genes in GSE93376

At the same time, we verified the differential expression of seven pyroptosis-related genes (Aim2, Casp8, Gsdmd, Naip2, Naip5, Naip6 and Trem2) based on GSE93376. It was revealed that Casp8, Gsdmd and Trem2 were significantly up-regulated in MCAO mice, while no significant differences were observed for the remaining four genes (Fig. 5).

Figure 5 Verification of seven pyroptosis-related genes in GSE93376.

Differentially expressed pyroptosis-related genes were up-regulated in MCAO mice

In addition, mouse MCAO models were constructed to verify the expression of Casp8, Gsdmd and Trem2. The evaluation of neurologic function and TTC staining results showed that the mouse MCAO model was successfully built (Figs. 6A and 6B). Moreover, the inflammatory factor TNF-α as well as the pyroptosis-related proteins were both increased in the MCAO group (Figs. 6C and 6D). Finally, Western blot and qRT-PCR analysis both showed that the expression levels of Casp8, Gsdmd and Trem2 were significantly higher in the MCAO group than those in the Sham group (Figs. 6E and 6F). Taken together, pyroptosis-related genes Casp8, Gsdmd, and Trem2 exhibited higher expression levels in the brain tissues of mice with cerebral infarction.

Figure 6 Experimental verification of pyroptosis-realated genes in mice.

(A) The evaluation of neurologic function in mice, ***P < 0.001; (B) TTC staining results for the brain tissues of mice; (C) detection of the protein expressions of TNF-α by ELISA, ***P < 0.001; (D) detection of the protein expressions of Cleaved caspase-1 and NLRP3 by Western blot; (E) detection of the protein expressions of Casp8, Gsdmd and Trem2 by Western blot; (F) detection of the gene expressions of Casp8, Gsdmd and Trem2 by qRT-PCR, **P < 0.01, ***P < 0.001.

Differentially expressed pyroptosis-related genes were up-regulated in BV2 cells

Previous studies have already identified Casp8 and Gsdmd as marker proteins for pyroptosis, so we chose Trem2 for follow-up studies. Trem2 has been shown to be primarily expressed in microglia, which are known to play an important role in ameliorating the progression of cerebral infarction. In vitro, BV2 microglia were subjected to OGD treatment, resulting in increased expression of Trem2 (Figs. 7A and 7B). We found a significant increase in Trem2 in the OGD group, consistent with the results obtained from both previous bioinformatics analyses and animal–level validations. Compared with the Control group, BV2 cells in the OGD group exhibited shortened processes, round or rod-shaped cell morphology, and enlarged cell bodies, indicative of microglial activation (Fig. 7C). Additionally, the BV2-secreted inflammatory factors IL-1β, TNF-α, and IL-6 increased (Fig. 7D), along with a significant increase in pyroptosis-related proteins in the OGD group (Fig. 7E). In both the OGD+si-NC group and the OGD+si-Trem2 group (Fig. 7C), BV2 activation was observed. Compared with the OGD+si-NC group, the OGD+si-Trem2 group demonstrated significantly increased secretion of inflammatory factors IL-1β, TNF-α, and IL-6, as well as pyroptosis-related proteins (Fig. 7D and 7E). In OGD-treated BV2 cells, both the expression of Trem2 and the levels of pyroptosis increased. Upon knockdown of Trem2 expression in OGD-treated BV2 cells, the levels of pyroptosis further increased. Therefore, we conclude that Trem2 is a protective factor that regulates pyroptosis, thus impacting the progression of cerebral infarction.

Figure 7 Differentially expressed pyroptosis-related genes were up-regulated in BV2 cells.

(A) Detection of the protein expressions of Trem2; (B) detection of the gene expressions of Trem2, **P < 0.01; (C) Cell morphology; (D) detection of the protein expressions of IL-1β, TNF-α, and IL-6, **P < 0.01; (E) Detection of the protein expressions of Cleaved caspase-1, NLRP3 and Gsdmd.

Discussion

Cerebral infarction can cause serious damage to the body (Maida et al., 2020). Thrombolytic therapy is the main treatment for cerebral infarction, but there are known defects in this kind of therapy (Zhao et al., 2016; Zhang et al., 2018). Therefore, our research aimed to explore new therapeutic targets for cerebral infarction through bioinformatics technology.

Pyroptosis is mainly mediated by non-classical and classical inflammasome pathways, and the classical inflammasome pathway activates and aggravates the brain injury in cerebral infarction (Lu et al., 2020; Wang et al., 2020; De Vasconcelos & Lamkanfi, 2020; Shen et al., 2021). However, inhibition of inflammasome and downstream molecules can reduce such brain injury and play a protective role (Gou et al., 2021). At the same time, the classical inflammasome pathway mediates the cell damage in cerebral infarction, so pyroptosis is a potential target for the intervention of cerebral infarction (Poh et al., 2019; Liu et al., 2021a). The inhibition of caspase-1 expression was demonstrated to have a neuroprotective effect on MCAO injury (Li et al., 2019). Caspase-3 mediates neuronal pyroptosis in cerebral infarction (Love, 2003). Pyroptosis plays a key role in cerebral infarction. Therefore, the current study explored the mechanism of pyroptosis in cerebral infarction. Pyroptosis-related genes Casp8, Gsdmd and Trem2 were screened out in cerebral infarction by bioinformatics, and they were verified to be highly expressed in cerebral infarction. In addition, we constructed a mouse MCAO model through animal experiments, and confirmed the high expression of pyroptosis-related genes Casp8, Gsdmd and Trem2 in the brain tissues of mice with cerebral infarction. These results not only confirmed the accuracy of bioinformatics results, but also revealed the potentially significant roles of Casp8, Gsdmd and Trem2 in cerebral infarction.

Casp8 is a cysteine-containing aspartate-specific proteases that functions in initiating caspase in the pathway of inducing cell death (Ghavami et al., 2009; Liu, Vetreno & Crews, 2021b). In addition, Casp8 is an exogenous coactivator of apoptosis that activates calpain I (Sarhan et al., 2018; Fritsch et al., 2019). The N-terminal domain of Gsdme has been revealed to promote cell pyroptosis by activating calprotease I or Casp8 (Zhou et al., 2018; Lee et al., 2018). In vitro studies have indicated that Casp8 can induce cell production independent of casp8 protease activity (Hartwig et al., 2017; Henry & Martin, 2017). High expression of Casp8 causes embryonic death in mice by inducing apoptosis and pyroptosis (Wang & Kanneganti, 2021). In addition, Casp8 is a molecular switch to control pyroptosis, necrosis and apoptosis, especially in embryonic development and growth (Fritsch et al., 2019). Therefore, Casp8 can be a potential therapeutic target for patients with cerebral infarction.

Gsdmd belongs to the Gasdermin protein family, which also includes Gsdmb, Gsdmcd, and Gsdma of which Gsdmd contains two domains: C-terminal and N-terminal (Shi et al., 2015; Kayagaki et al., 2015; Sborgi et al., 2016; Ding et al., 2016; Aglietti & Dueber, 2017). In addition, when Gsdmd is cleaved by the Caspase protein, which leads to the destruction of cell membrane, changes in cell osmotic pressure, destruction of cell membrane function, and ultimately the occurrence of pyroptosis (Shen et al., 2021; Sborgi et al., 2016; Ding et al., 2016; Aglietti & Dueber, 2017). It found that intestinal ischemia injury by knocking down Gsdmd (Jia et al., 2020). Gsdmd is a common substance released during pyroptosis and by inflammatory cells (Rello et al., 2017; Hu et al., 2020). The above conclusions indicate that Gsdmd is a molecular marker of pyroptosis, which is consistent with the experimental results of this research.

Trem2 has been detected in myeloid, platelet and endothelial cells (Grant et al., 1990; Cella et al., 2003; Turnbull et al., 2006; Jiang et al., 2013). Trem2 protein is a single transmembrane receptor belonging to the immunoglobulin (Painter et al., 2015). In addition, Trem2 binds to transmembrane signal receptor DAP12 (Kawabori et al., 2013; Sieber et al., 2013; Jiang et al., 2014). Trem2 improves the inflammatory response in vitro by inhibiting the secretion of cytokines by microglia (Takahashi, Rochford & Neumann, 2005; Takahashi et al., 2007). Loss of Trem2 function triggers a chronic inflammatory response that exacerbates Alzheimer’s disease (Abduljaleel et al., 2014). Therefore, Trem2 has become a new clinical therapeutic target for cerebrovascular diseases (Wu et al., 2017). At the same time, Trem2 has been revealed to protect against inflammation (Qu et al., 2018). The results of this study suggest that Trem2 can be a therapeutic target for patients with cerebral infarction. Based on our results, Trem2 has been shown to regulate microglial pyroptosis. In sepsis, Trem2 has been identified as an inhibitor of pyroptosis in bone marrow-derived macrophages (Yang et al., 2021), suggesting its involvement involved in this process. Moreover, Buyang Huanwu decoction (She et al., 2019) and edaravone dexborneol (Hu et al., 2022) was found to play a neuroprotective role by inhibiting neuronal pyroptosis after brain ischemia/reperfusion injury.

In conclusion, pyroptosis-related genes Casp8, Gsdmd, and Trem2 were screened out and validated by bioinformatics methods. In addition, the high expression of Casp8, Gsdmd and Trem2 in mice with cerebral infarction was confirmed by animal experiments. At the cellular level, Trem2 was confirmed to regulate the pyroptosis of BV2 cells, thereby affecting the progression of cerebral infarction. We will further verify the mechanism of these three genes in the progression of cerebral infarction at the animal level, and collect as many clinical samples as possible to confirm that these three genes can be used to predict the progression of cerebral infarction. This study highlighted the regulatory role of Casp8, Gsdmd and Trem2 in pyroptosis, underscoring their potential impact on cerebral infarction. These genes may serve as new therapeutic targets for patients with cerebral infarction.

Supplemental Information

Supplemental Information 1 Raw data

Click here for additional data file.

Supplemental Information 2 GO 0070269 Dataset Genes

Click here for additional data file.

Supplemental Information 3 ARRIVE 2.0 Checklist

Click here for additional data file.

Supplemental Information 4 MIQE Checklist

Click here for additional data file.

Additional Information and Declarations

Competing Interests

Author Contributions

Animal Ethics

Data Availability

The authors declare there are no competing interests.

Shunli Liang conceived and designed the experiments, authored or reviewed drafts of the article, and approved the final draft.

Linsheng Xu performed the experiments, analyzed the data, prepared figures and/or tables, and approved the final draft.

Xilin Xin performed the experiments, prepared figures and/or tables, and approved the final draft.

Rongbo Zhang performed the experiments, prepared figures and/or tables, and approved the final draft.

You Wu conceived and designed the experiments, authored or reviewed drafts of the article, and approved the final draft.

The following information was supplied relating to ethical approvals (i.e., approving body and any reference numbers):

The Institutional Animal Care and Use Committee (IACUC), Zhejiang Provincial Committee for Laboratory Animal (ZJCLA)

The following information was supplied regarding data availability:

The raw data is available in the Supplemental Files.

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
