# Peer review of "Study on pyroptosis-related genes Casp8, Gsdmd and Trem2 in mice with cerebral infarction"

_PeerJ, doi:10.7717/peerj.16818_

## Round 0.1 · original submission · Major Revisions

Please address the concerns of reviewers and revise the manuscript accordingly.

Reviewer 1 ·

Basic reporting

The study is important for science.

The introduction is clear and understandable.


Please, pay attention to formatting and writing.

Experimental design

Line 102 - Who is the control group? sham operation group? Please, describe the control.
Line 106/104 -Describes better how the selection of animals was carried out using scores from 1 to 4.
Line 121 – And the sham operation group?
Line 150/152 - Figure 1 (A; B,C and D) does not identify the experimental groups. Please, identify each figure with its corresponding group.
Line 159/165 - Was this result based only on the treated group (MCAO mice)? and the control?
Line 167/170 - Please, identify each figure with its corresponding group.

How was each score statistically evaluated?
(1,2, 3, 4)

Validity of the findings

Line 72 - The article suggests new therapeutic targets, not new treatments.
Please add the fundings, author contributions and conflicts of interest

Additional comments

The authors did not make it clear how the neurological examination was performed.

It is important to explain to readers how the statistics were performed according to the animal score (neurological examination).

The statistical analysis is still worrying.

·

Basic reporting

This study demonstrates high expression of pyroptosis related genes Casp8, Gsdmd and Trem2 in mice with cerebral infarction. Although this study is interesting and important in targeted therapy for cerebral infarction, the study can be improved by more complete characterization of mice and addressing few key issues. The following suggestions are provided to help strengthen this work.
1. It is important to justify the rationale behind selecting mice aged 3 and 18 months from the datasets for the study and generation of cerebral infarction model.
2. There are other ways to confirm the neurological score. For example, it can be done by checking gait, retraction of legs etc.
3. Please address the degree of infarction in these mice models. Is there any method to verify this?
4. It is ideal to include a broad panel of genes in RTPCR and western blot analysis although 3 genes were found to be upregulated in enrichment analysis.
5. The study does not provide compelling evidence for the pyroptosis association. More experiments in mice and in-vitro are needed to validate this. In addition, the mechanistic insights of the molecular pathways involved in cerebral infarction in these mice is not addressed.

Experimental design

.

Validity of the findings

.

Reviewer 3 ·

Basic reporting

The study found that Casp8, Gsdmd, and Trem2 were significantly up-regulated in cerebral infarction mice, offering potential therapeutic targets.

Experimental design

The study utilizes bioinformatics analysis to identify possible therapeutic targets. However, it's important to note that the clinical relevance of these findings to human patients with cerebral infarction may not be directly transferable. Also, the study mentions the construction of a mice MCAO model, but it is important to note that results in animal models may not fully reflect the complexities of human cerebral infarction.

Validity of the findings

For the result section, while this section mentions the number of differentially expressed genes (DEGs) and their functional enrichment, it lacks specific details such as fold changes, statistical significance, etc., those details are important to assess the strength of the findings.

For the discussion section, the study identifies the high expression of Casp8, Gsdmd, and Trem2 in cerebral infarction, but it doesn't provide detailed insights into the mechanisms. Understanding the exact pathways and interactions is essential for the development of targeted therapies.

The discussion does not compare the newly identified therapeutic targets (Casp8, Gsdmd, Trem2) with existing treatment options or potential alternatives. Comparative analysis would provide a more comprehensive view of their potential clinical significance.

---

## Round 0.2 · Minor Revisions

Please address the remaining issues pointed out by the reviewers and amend the manuscript accordingly.

Reviewer 1 ·

Basic reporting

Line. 67 - The study evaluates new targets for patients with cerebral infarction.
However, the authors incorrectly describe: "we provide new treatments for patients with cerebral infarction". Please modify.

Experimental design

Line. 79 - please, identify Venn diagram in the text (figure number)
Line. 82 - ClusterProfiler software: please, describe and add reference.

The authors did not explain the total number of animals used in the experiment. For example:

Data Acquisition
GSE137482 and GSE93376 were downloaded from the GEO. RNA-SEQ of the brain
tissues from 24 mice were included in GSE137482: including 12 mice in the middle cerebral
artery occlusion model group (MCAO group) and 12 mice in the sham operation group (Sham
74 group). There were the brain tissues of 6 mice in GSE93376: 3 MCAO models and 3 sham
75 operation samples. ( i.e. 12 MCAO and 12 Sham)

On the other hand, the authors when describing the Establishment of mouse MCAO Model, report:
Twenty C57BL/6J mice (23-25 g), aged 8-10 weeks (Nanjing Institute of Biomedical
95 Sciences, Nanjing University), were selected and divided into two groups according to the
96 random number table method: the MCAO model group and the sham operation group, with 6
97 mice in each group. ( i.e. 20 MCAO)
How many animals were used? 20 or 24
How many animals were used in SHAM? Was there 42 animals? Please describe describe in detail.
Line. 122 - What were the normal conditions?
Line. 129 - how many mices were euthanized?
Line. 151 - Please, describe.

Validity of the findings

Please, I suggest that the statistics are performed using another statistical program (SAS (Statistical Analysis System or SPSS Statistics). Graphpad prism 9.0 software is an excellent program just to produce graphs.
What were the parametric and nonparametric data?

·

Basic reporting

Good

Experimental design

Revised.

Validity of the findings

Addressed.

Reviewer 3 ·

Basic reporting

no comment

Experimental design

no comment

Validity of the findings

For the discussion section, while the acknowledgment of the need for further investigation into the pathways involving Trem2 and its potential impact on pyroptosis is appreciated, the authors' response lack specificity. It would be impactful to outline the specific methodologies that could be pursued in future studies to address this gap in understanding. Please providing suggestions or hypotheses on potential mechanisms would enhance the depth of the response.

Additional comments

no comment

---

## Round 0.3 · accepted · Accept

Thank you for addressing the remaining concerns of the reviewers and for making corresponding changes. Your revised manuscript is acceptable now.